# Do people have an ethical obligation to share their health information? Comparing narratives of altruism and health information sharing in a nationally representative sample

Minakshi Raj[1]*, Raymond De Vries[2], Paige Nong[3], Sharon L. R. Kardia[4], Jodyn E. Platt[5]

1 Department of Kinesiology and Community Health, University of Illinois at Urbana Champaign, Champaign, IL, United States of America, 2 Center for Bioethics and Social Sciences in Medicine, University of Michigan, Ann Arbor, MI, United States of America, 3 Department of Health Management and Policy, University of Michigan School of Public Health, Ann Arbor, MI, United States of America, 4 Department of Epidemiology, University of Michigan School of Public Health, Ann Arbor, MI, United States of America, 5 Department of Learning Health Sciences, University of Michigan Medical School, Ann Arbor, MI, United States of America

* miraj@umich.edu

**Data Availability Statement:** All relevant data are within the manuscript and its Supporting Information files.

## Abstract

### Background

With the emergence of new health information technologies, health information can be shared across networks, with or without patients' awareness and/or their consent. It is often argued that there can be an ethical obligation to participate in biomedical research, motivated by altruism, particularly when risks are low. In this study, we explore whether altruism contributes to the belief that there is an ethical obligation to share information about one's health as well as how other health care experiences, perceptions, and concerns might be related to belief in such an obligation.

### Methods

We conducted an online survey using the National Opinion Research Center's (NORC) probability-based, nationally representative sample of U.S. adults. Our final analytic sample included complete responses from 2069 participants. We used multivariable logistic regression to examine how altruism, together with other knowledge, attitudes, and experiences contribute to the belief in an ethical obligation to allow health information to be used for research.

### Results

We find in multivariable regression that general altruism is associated with a higher likelihood of belief in an ethical obligation to allow one's health information to be used for research (OR = 1.22, SE = 0.14, p = 0.078). Trust in the health system and in care providers are both associated with a significantly higher likelihood of believing there is an ethical

**Funding:** SLRK and JEP received NIH grant 5R01 CA214829-03 (The Lifecycle of Health Data: Policies and Practices) URL: https://projectreporter.nih.gov/project_info_description.cfm?aid=9852993&icde=51752022 The funders had no role in study design, data collection and analysis, decision to publish, or preparation of the manuscript.

**Competing interests:** The authors have declared that no competing interests exist.

obligation to allow health information to be used (OR = 1.48, SE = 0.76, p<0.001; OR = 1.58, SE = 0.26, p<0.01, respectively).

## Conclusions

Belief that there is an ethical obligation to allow one's health information to be used for research is shaped by altruism and by one's experience with, and perceptions of, health care and by general concerns about the use of personal information. Altruism cannot be assumed and researchers must recognize the ways encounters with the health care system influence (un)willingness to share one's health information.

## Introduction

Recruitment of human subjects is fundamental to the validity and success of health research. The decision to participate in health research has been discussed from two perspectives over recent decades. The first frames the decision to participate in research as a moral good, but not a requirement. That is, one who goes beyond the call of duty and participates is viewed positively, but one who does not participate is not considered negatively [1–5]. The second, and more predominant view in recent years, regards participation in research as an obligation because (a) participating is good for society, (b) society benefits from others who participate in research, and (c) it is a public good or even a positive externality [6–14]. In their ethical framework for learning health systems, for example, Faden and colleagues (2013) assert that patients have an obligation to participate in research in order to contribute to improving the quality of the healthcare system and to improve knowledge [15]. While these two perspectives are discordant with respect to whether participation in research is a *voluntary* moral good or an *obligation*, they both see enrolling in a research protocol as an act of altruism. In other words, these perspectives agree that people have an obligation to contribute to research, such as, by sharing health information, if the research poses low risks to an individual and has potential benefits to everyone.

Altruism is defined as behaving in a way that benefits another, prioritizing selflessness and helpfulness in order to make a positive contribution to another person or to the larger society [16]. Altruism involves sacrificing one's wellbeing for the wellbeing of others without expectation of personal compensation or benefit [17]. This orientation is an essential aspect of being a research participant and for this reason those who agree to participate in biomedical research must be informed, before enrolling, that while their participation may not benefit them personally, the study is designed to contribute to the advancement of treatments to help others [18]. As such, participation in research is one instance of altruism—selfless behavior that promotes the wellbeing of others.

The nature and risks associated with research participation are highly varied. A research participant may be asked to do a cognitive task, sacrifice privacy for a sleep study, or contribute a biospecimen for the study of disease. With the emergence of new technologies and the expansion of possibilities for sharing and linking health information such as through learning health systems, the forms of research participation continue to evolve [19]. Research using health information introduces even greater opacity with regard to the meaning of research participation and the obligation to enroll in research. Active participation in research positions subjects proximally to the generation of new knowledge, while sharing health information is more distanced. "Participants" may not even be aware of their role in generating knowledge or the procedures by which data are translated into knowledge. But this does not mean these

studies are free from risk to participants. Sharing and linking health data threaten confidentiality and these data can result in harm to individuals [20]. Furthermore, one's religious, cultural, and moral beliefs may lead to concerns about how these data are used [21]. Participants, despite wanting to contribute to public good, grapple with the risks they may individually face or perceive that others face [20]. They may not even be aware of the possibility that their health information is being used beyond the boundaries of their consent. In the context of declining trust in the medical profession, the health system, and the research enterprise, there is a need to fully consider how the (mis)use of health information can accelerate the breakdown of relationships between patients and caregivers/researchers [22].

Participating in research may sometimes involve giving up a part of oneself—i.e. making oneself vulnerable—and entrusting researchers to use it for the betterment of society and not take advantage of or harm subjects as was the case in the Public Health Service's Tuskegee Syphilis Study [23, 24]. The history of the research enterprise is rife with failures to protect human subjects at the point of research or in handling health information. The now well-known case of Henrietta Lacks is a prominent example of the use of biospecimens without the knowledge or consent of a patient [25]. The case of the Havasupai Tribe in Arizona presents another instance in which research subjects were not appropriately informed of how their DNA would be used in the future [26]. These cases represent a subset of instances that may threaten the link between altruism and participating in research, diminishing the felt obligation to participate in research. Furthermore, individual experiences of discrimination may lead to a lack of confidence in the health care system to be trustworthy in holding up the implicit obligation to 'do no harm.' Similarly, the perception that discrimination occurs in the healthcare system–such as discrimination based on race/ethnicity income—may be related to beliefs about sharing one's health information. For instance, some may perceive that discrimination occurs in the health system and may believe that they have an obligation to share their health information for research in order to help others; in contrast, others who perceive discrimination occurs may not believe they have an obligation to allow their health information to be used for research because of a mistrust in the intentions of the research enterprise.

In our earlier study, we demonstrated that while willingness to be part of multi-user data networks is low, altruism is a significant predictor of the willingness to participate in these types of networks [27]. However, little is known about how altruism contributes to the specific belief that people have an ethical obligation to allow their *health information* to be used for research. The extent to which altruism contributes to the belief that there is an ethical obligation to allow health information to be used for research remains unclear. Further, there is a need to understand characteristics–other than altruism–of those who do *not* believe there is an ethical obligation to allow health information to be used for research. Are there certain types of experiences, such as the experience of discrimination, that inhibit people from believing that they should share their health information for research? Does the *perception* that discrimination occurs in the health care system influence willingness to share one's information? Do failures in the protection of personal information (e.g. in the case of Memorial Sloan Kettering, where patient data was used without their knowledge for a profit-driven venture, Paige.AI) have broader implications for information sharing, including health information [28]? Is the nature of the information (e.g., identified or de-identified) related to beliefs about information sharing for research?

In the current study, we examine these questions, looking at how different beliefs (e.g., altruism), experiences and perceptions about the health care system (e.g., discrimination, trust), concerns (e.g., about previous breaches) and comfort with information sharing relate to the belief that people have an ethical obligation to participate in research.

## Materials and methods

### Sample

We used the National Opinion Research Center's (NORC) probability-based, nationally representative sample of U.S. adults to conduct an online survey in May 2019. This protocol was approved by the IRB at the University of Michigan. Prior to data collection NORC pre-tested the survey via interviews with 115 panelists. A total of 2,157 individuals participated and completed the survey (66% complete response rate). We oversampled African Americans, Latinos and individuals earning less than 200% of the federal poverty level annually. The final analytic sample of 2,069 resulted from inclusion of participants with complete information for variables used in our analysis. This study was given permission with exemption by the University of Michigan Institutional Review Board (HUM00161912). A written consent form was included in the survey.

### Measures

Our main outcome measure was *altruism as it relates to research* as represented in the response to the following prompt: *People have an ethical obligation to allow health information to be used for research*. Respondents answered on a four-point scale ranging from "Not true" to "Very true". In our final model, a logistic regression analysis identifying factors associated with the belief that people have an ethical obligation to allow health information be used for research, we created a binary variable indicating whether there was *some* obligation (i.e., the statement is somewhat true, fairly true, or very true) or no obligation (not true).

Our independent variables included (a) demographic characteristics, (b) health care system experiences, and (c) health care system perceptions, (d) health system and provider trust, (e) concerns about the security of personal information, (f) comfort and preferences related to health information sharing, and (g) general altruism.

**Demographic characteristics.** We measured age categorized into two groups (<50 and 50+), self-reported sex (male or female), race/ethnicity (non-Hispanic white, non-Hispanic black, non-Hispanic other, and Hispanic), education (less than high school, high school graduate, some college, or bachelor's degree or higher), income (less than $50,000/year or $50,000/year or more). In addition, we measured self-reported health status on a five-point scale ranging from poor to excellent.

**Experiences with the health care system.** We assessed three types of experiences with the health care system. We asked participants whether they had ever been discriminated against, hassled, or made to feel inferior while getting medical care [29]. Participants were also asked to respond to the following prompts: *My healthcare system treats me with kindness*, and *My healthcare system treats me fairly* on a four-point scale ranging from "Not true" to "Very true".

**Perceptions of the health care system.** Participants responded to the following two prompts: *The organizations that have my health information and share it treat everyone the same regardless of their race or ethnicity*, and, *The organizations that have my health information and share it treat everyone the same regardless of their income* on a four-point scale ranging from "Not true" to "Very true".

**System and provider trust.** To measure participant trust in the health care system, we asked participants to respond on a four-point scale ranging from "Not true" to "Very true" to four items measuring *integrity*, three items measuring *competency*, and three items measuring *globalized trust*. For *integrity*, participants were asked to respond to whether the organizations that have their health information and share it: *Try hard to be fair in dealing with others*, *Would try to hide a serious mistake*, *Tell me how my health information is used*, and *Would*

*never mislead me about how my health information is used*. For *competency*, we asked participants whether the organizations that have their health information and share it: *Are not good at their jobs*, *Have specialized capabilities that can promote innovation in health*, and *Can use large amounts of data to improve patient care*. For *globalized trust*, participants responded to whether the organizations that have their health information and share it: *Can be trusted to use my health information responsibly*, *Think about what is best for me*, and *Act in an ethical manner*. We then created a system trust index as the sum of each participant's responses to the questions divided by the number of questions answered, resulting in a continuous variable with values ranging from 3 to 12 [30].

To measure provider trust, we asked participants to use the same four point scale to reflect on health care providers–including people such as doctors and nurses who provide medical treatment–in response to the following prompts: *Health care providers care most about making money for themselves*; *I trust health care providers to use my health information responsibly*; *Health care providers do not care about helping people like me*; *Disclose their conflicts of interest*; and, *All things considered, health care providers in the U.S. can be trusted*. We created an index of provider trust using the same process described previously, resulting in a continuous variable with values ranging from 1 to 4.

**Concerns about security of personal information.** We asked participants to indicate on a four-point scale from "Not at all concerned" to "Very concerned", how concerned they are about the following incidents resulting in compromised personal information: *Facebook sharing information with Cambridge Analytica for political purposes*, and, *Sloan Kettering hospital executives using hospital data for their own startup company*.

**Comfort and preferences with health information sharing.** We asked participants to respond to four questions on a four- point scale from "Not true" to "Very true" to assess their comfort and preferences related to sharing of de-identified and identified health information: *I am comfortable with university researchers using my de-identified health information*, *I would like to be notified if university researchers will use my de-identified health information*, *I am comfortable with university researchers using my identified health information*, and, *I would like to be notified if university researchers will use my identified health information*.

We provided the following definitions to participants: (a) *Health information* includes information includes information about you and your medical treatment history including diagnoses, medications, treatment plans, immunization dates, allergies, radiology images, and laboratory and test results; (b) *De-identified health information* means that "identifying information" about you is *removed* from your health information; and (c) *Identified health information* includes things like your name, address, date of birth, etc.

**General altruism.** Items measuring general altruism were drawn from the National Election Survey, and we asked participants to respond to four questions on a four-point scale of "Not true" to "Very true": *I find ways to help others less fortunate than myself*; *The dignity and well-being of all should be the most important concern in any society*; *One of the problems of today's society is that people are often not kind enough*; and *All people who are unable to provide for their own needs should be helped by others* [31]. We then created an index as the sum of the participant's responses to those survey questions divided by the number of questions answered. Development of indices for altruism based on principal component analysis and initial assessments of Cronbach's alpha (a = 0.69) are described elsewhere [32].

## Analysis

We began with descriptive analyses of the full sample. We then conducted bivariable analysis to examine the relationship between demographic factors, experiences, concerns, preferences,

trust, and general altruism as they each relate to the belief in an ethical obligation to allow health information to be used for research. Then, we conducted multivariable logistic regression to examine the relationship between general altruism and the belief in an ethical obligation to allow health information to be used for research, controlling for demographic factors, experiences, perceptions, concerns, preferences, and trust. We used survey weights as estimated by NORC and used Stata Version 15.1 for analyses.

## Results

The mean age of participants was 49.6 years and the sample reflected the racial composition of the U.S. Just over half of respondents were female, and 52% had an annual income of less than $50,000 (Table 1). Participants were relatively healthy with almost three-quarters responding that their health is good, very good, or excellent.

### Health care experiences and perceptions

About one-fifth (21.3%) of respondents reported having an experience of discrimination in the health care system, and nearly three-quarters reported that it is *fairly true* or *very true* that the health care system treats them with fairness (70.9%) or with kindness (72.1%). More than half of respondents perceive that discrimination based on race/ethnicity (57.3%) or income (62.9%) occurs in the health care system *sometimes* or *often*. Mean trust in the health care system was 7.08 (sd = 1.9) on a scale from 3 to 12 while mean trust in health care providers was 2.21 (sd = 0.5) on a scale from 1 to 4.

**Table 1. Demographic characteristics of the sample (n = 2069).**

|  | Full sample n (%) |
|---|---|
| Age (years) |  |
| mean (sd) | 49.6 (16.3) |
| Age categories |  |
| < 50 | 1024 (49.5) |
| 50+ | 1045 (50.5) |
| Education |  |
| No high school diploma | 80 (3.9) |
| High school equivalent | 367 (17.7) |
| Some college | 956 (46.2) |
| BA or above | 666 (32.2) |
| Race/Ethnicity |  |
| White, non-Hispanic | 1203 (58.1) |
| Black, non-Hispanic | 325 (15.7) |
| Other, non-Hispanic | 136 (6.6) |
| Hispanic | 405 (19.6) |
| Sex |  |
| Female | 1054 (50.9) |
| Male | 1015 (49.1) |
| Income |  |
| < $50,000 | 1078 (52.1) |
| $50,000 + | 991 (48.9) |
| Health Status (Range poor = 1—excellent = 5) |  |
| Mean (sd) | 3.08 (0.93) |

## Concerns and preferences regarding information sharing

About two-thirds of the sample reported being *fairly* or *very* concerned about the Facebook and Memorial Sloan Kettering Cancer Center (MSKCC) breaches of trust.

Mean responses regarding comfort (mean = 2.92, sd = 1.1) and preference for notification (mean = 2.77, sd = 1.2) were similar, while mean comfort with researchers using identified health information was low (mean = 1.95, sd = 1.1) and preference for being notified if researchers are using identified health information was high (mean = 3.43, sd = 1.0).

## Beliefs about allowing information to be used for research

Roughly half (50.7%) of the analytic sample responded that it is *not true* that people have an ethical obligation to allow their health information to be used for research. On a four-point continuous scale, mean altruism was 2.97 (sd = 0.7) (Table 2).

Higher altruism was associated with a greater likelihood of responding that people have an ethical obligation to allow their health information to be used for research (OR = 1.22, SE = 0.14) (Table 3). When compared to respondents younger than age 50, participants aged 50 and older were significantly less likely to report that people have an ethical obligation to allow their health information to be used for research (50+, OR = 0.69, SE = 0.10, p<0.05).

Participants who had experienced discrimination were less likely to respond that people have an ethical obligation to allow their health information to be used for research than those who had not experienced discrimination (OR = 0.87, SE = 0.15). Respondents who perceive discrimination occurs in the health system based on race/ethnicity *sometimes* or *often* (OR = 1.54, SE = 0.33, p<0.05) were significantly more likely to respond that there is an obligation to allow health information to be used compared with those who do not perceive that this discrimination occurs. System trust and provider trust were both associated with a significantly higher likelihood of responding there is an ethical obligation to allow health information use (OR = 1.48, SE = 0.76, p<0.001 and OR = 1.58, SE = 0.26, p<0.01 respectively).

Concerns about the Facebook breach and Memorial Sloan Kettering Cancer Center's relationship with Paige.AI were both associated with a lower likelihood of believing there is an ethical obligation to allow health information to be used (OR = 0.83, SE = 0.14; OR = 0.98, SE = 0.16, respectively).

Being more comfortable with researchers using de-identified or identified health information were both associated with a higher likelihood of responding that people have an ethical obligation to allow health information use for research (OR = 1.01, SE = 0.79 and OR = 1.48, SE = 0.11, respectively) and this relationship was significant for identified health information. Wanting to be notified if researchers are using de-identified or identified health information were both associated with a lower likelihood of reporting an ethical obligation to allow health information to be used for research purposes (OR = 0.95, SE = 0.06 and OR = 0.77, SE = 0.06 respectively) and again, the relationship was significant for identified health information.

## Discussion

In this study, we used a US sample to examine the relationship between altruism and the ethical obligation to allow health information to be used for research. Our finding that general altruism is associated with a higher likelihood of believing there is an ethical obligation to allow health information to be used for research suggests that altruism may contribute to this belief; however other characteristics such as trust in the system and in providers may also contribute to the belief that there is an ethical obligation to allow health information use for research. In particular, we also found that different types of experiences, perceptions, concerns, and preferences regarding the health care system and health information sharing are related to

**Table 2. Descriptive summary of outcomes and predictors of interest (n = 2069).**

| | Frequency (%) |
|---|---|
| Health care experiences | |
| No experience of discrimination | 1629 (78.7) |
| Experience of discrimination | 440 (21.3) |
| *Not true or somewhat true* that the health system treats me fairly | 603 (29.1) |
| *Fairly or very true* that the health system treats me fairly | 1466 (70.9) |
| *Not true or somewhat true* that the health system treats me with kindness | 577 (27.9) |
| *Fairly or very true* that the health system treats me with kindness | 1492 (72.1) |
| Health system perceptions | |
| Discrimination based on race/ethnicity occurs *never or rarely* | 883 (42.7) |
| Discrimination based on race/ethnicity occurs *sometimes or often* | 1186 (57.3) |
| Discrimination based on income occurs *never or rarely* | 767 (37.1) |
| Discrimination based on income occurs *sometimes or often* | 1302 (62.9) |
| System trust (index; range 3–12) | |
| Mean (sd) | 7.08 (1.9) |
| Provider trust (index; range 1–4) | |
| Mean (sd) | 2.21 (0.5) |
| Concerns about information sharing | |
| *Not at all or somewhat* concerned about Facebook breach | 628 (30.3) |
| *Fairly or very* concerned about Facebook breach | 1441 (69.7) |
| *Not at all or somewhat* concerned about MSKCC breach | 707 (34.2) |
| *Fairly or very* concerned about MSKCC breach | 1362 (65.8) |
| Information sharing comfort and preferences | |
| Comfort with researchers using de-identified health information[1] | |
| Mean (sd) | 2.92 (1.1) |
| Would like to be notified if researchers are using de-identified health information[1] | |
| Mean (sd) | 2.77 (1.2) |
| Comfort with researchers using identified health information[1] | |
| Mean (sd) | 1.95 (1.1) |
| Would like to be notified if researchers are using identified health information[1] | |
| Mean (sd) | 3.43 (1.0) |
| General Altruism (index) | |
| Mean (sd) | 2.97 (0.7) |
| *Not true* that people have an ethical obligation to allow their health information to be used for research | 1045 (50.5) |
| *Somewhat*, *fairly*, *or very true* that people have an ethical obligation to allow their health information to be used for research | 1024 (49.5) |

Note

[1]Variables are continuous on a four-point scale from *Not true* to *Very true*

the belief that people have an ethical obligation to allow health information to be used for research. For instance, being 50 or older, perceiving that discrimination based on race/ethnicity occurs in the health system, higher self-reported health status, the comfort with researchers' use of identified health information and the preference for being notified about researchers' use of identified health information, are all associated with a lower likelihood of believing people have an ethical obligation to allow their health information to be used for research. That comfort with researchers' use of identified health information is associated with a lower likelihood of believing there is an ethical obligation to allow health information use for research

**Table 3. Bivariable and multivariable odds ratios relating demographic, experience, perception, concern, comfort and preference characteristics to the ethical obligation to allow health information to be used for research (n = 2069).**

| | Bivariable Odds Ratio (SE) | Multivariable Odds Ratio (SE) |
|---|---|---|
| *Demographic characteristics* | | |
| *Age Category* | | |
| < 50 | Ref | Ref |
| 50+ | 0.73 (0.09)* | 0.69 (0.10)* |
| *Sex* | | |
| Male | Ref | Ref |
| Female | 1.01 (0.12) | 1.25 (0.17) |
| *Race/Ethnicity* | | |
| White, NH | Ref | Ref |
| Black, NH | 1.00 (0.17) | 0.67 (0.14) |
| Other, NH | 1.45 (0.33) | 1.71 (0.45)* |
| Hispanic | 1.07 (0.19) | 0.77 (0.15) |
| *Education* | | |
| Less than high school | Ref | Ref |
| High school graduate | 0.74 (0.24) | 0.97 (0.32) |
| Some college | 0.54 (0.17)* | 0.68 (0.21) |
| BA or above | 0.56 (0.17) | 0.83 (0.27) |
| *Income* | | |
| Less than $50,000 | Ref | Ref |
| $50,000 or higher | 0.97 (0.12) | 1.23 (0.18) |
| Health status | | |
| *Range Poor (1) to Excellent (5)* | 0.84 (0.06)** | 0.78 (0.06)*** |
| *Health care experiences* | | |
| No experience of discrimination | Ref | Ref |
| Experience of discrimination | 1.13 (0.08) | 0.87 (0.15) |
| Not true or somewhat true that the health system treats me fairly | Ref | Ref |
| Fairly or very true that the health system treats me fairly | 1.18 (0.16) | 0.91 (0.21) |
| Not true or somewhat true that the health system treats me with kindness | Ref | Ref |
| Fairly or very true that the health system treats me with kindness | 1.22 (0.17) | 1.01 (0.23) |
| *Health system perceptions* | | |
| Discrimination based on race/ethnicity occurs never or rarely | Ref | Ref |
| Discrimination based on race/ethnicity occurs sometimes or often | 0.73 (0.09)* | 1.54 (0.33)* |
| Discrimination based on income occurs never or rarely | Ref | Ref |
| Discrimination based on income occurs sometimes or often | 0.65 (0.08)*** | 1.11 (0.22) |
| System trust (index) | 1.48 (0.06)*** | 1.48 (0.76)*** |
| Provider trust (index) | 2.48 (0.40)*** | 1.58 (0.26)** |
| *Concerns about information sharing* | | |
| Not at all or somewhat concerned about Facebook breach | Ref | Ref |
| Fairly or very concerned about Facebook breach | 0.72 (0.10)* | 0.83 (0.14) |
| Not at all or somewhat concerned about MSKCC breach | Ref | Ref |
| Fairly or very concerned about MSKCC breach | 0.76 (0.10)* | 0.98 (0.16) |
| *Information sharing comfort and preferences* | | |

(*Continued*)

**Table 3.** (Continued)

| | Bivariable Odds Ratio (SE) | Multivariable Odds Ratio (SE) |
|---|---|---|
| Comfort with researchers using de-identified health information[1] | 1.36 (0.86)*** | 1.01 (0.79) |
| Would like to be notified if researchers are using de-identified health information[1] | 0.89 (0.05)* | 0.95 (0.06) |
| Comfort with researchers using identified health information[1] | 1.89 (0.12)*** | 1.48 (0.11)*** |
| Would like to be notified if researchers are using identified health information[1] | 0.70 (0.05)*** | 0.77 (0.06)*** |
| General Altruism (index) | 1.35 (0.14)** | 1.22 (0.14) |
| Pseudo $R^2$ | - | 0.15 |

*Notes*: Population weights were applied

[1]Variables are continuous on a four-point scale from *Not true* to *Very true*

* $p < 0.05$

** $p < 0.01$

***$p < 0.001$

suggests that comfort does not necessitate a sentiment of obligation; this has implications for informed consent procedures wherein it is important to distinguish that while patients may agree and be comfortable with the use of their identified health information, knowledge about the specific situation or context of its use may still be important to patients.

In contrast, perceiving that the health system discriminates against people based on race/ethnicity, having higher system and provider trust, and being comfortable with researchers using de-identified health information are associated with a higher likelihood of believing people have an ethical obligation to allow their health information to be used for research. That the perception of discrimination is associated with the belief in an ethical obligation to allow health information to be used for research is surprising and requires further study. In our bivariable analyses, we found that the perception of discrimination was associated with a lower likelihood of believing there is an ethical obligation to allow health information to be used for research; this association was positive in multivariable analyses indicating a complex relationship with demographic and experiential characteristics such as one's own experience of discrimination. It is also possible that perceiving that discrimination occurs in the system is related to concern for others and a belief that allowing information to be used for research may ultimately help those who are experiencing discrimination.

Our variables measuring the experience and perception of discrimination both changed direction between bivariable and multivariable models, indicating that there may be interactions that should be investigated in future analyses but were limited by low power in the present analysis. For example, it is possible that the marginalization caused by discrimination contributes to a greater desire for agency and choice when it comes to allowing health information to be used. In their qualitative study, Mattis and colleagues (2009) found that participants' challenges—such as racism—*sparked* their commitment to selfless giving [33]. This commitment could be informed by concern or empathy for others and a desire to improve conditions for others who are having similar experiences. Understanding discrimination as an exposure that may limit participation in the health care system is particularly relevant in further exploration of this dynamic [34]. Further, it is possible that the opacity of health information sharing in combination with failures in the research enterprise and health care system generally, reinforce mistrust in the intent and outcome of research. As a result, allowing health

information to be used may be seen more as a burden or undue act of selflessness that has no benefit for the society of members of one's ethnic group.

Our results suggest that although concerns about the sharing of health information may be similar to concerns about the sharing of other types of personal information (e.g. Facebook breach), the belief that it is an ethical imperative to allow use of health information for research is distinct from the belief in an imperative to share or give to enhance societal wellbeing. To our knowledge, studies have not parsed the distinction between altruism and the ethical obligation to allow health information to be used for research, despite encouragement to consider the two in similar ways [15]. Participation in research and the sharing of health information are essential for continued advancement of the science that promotes health and wellbeing. However, the health system cannot expect this type of sacrifice from participants without strengthening policies and governance related to confidentiality, privacy, and health information sharing, building trust, and engaging with prospective and active participants in navigating their preferences regarding consent for use of their health information.

Even those individuals who want to participate in research or allow their health information to be used for research for the greater good should be told what this means. Providers should not take advantage of patients or participants who demonstrate altruism but may be unfairly subjected to undue risks or burdens that could consequently exacerbate disparities. Practitioners and researchers should ask participants to consider what types of health information they are comfortable with sharing, with whom, for how long, and to what extent. They should also have discussions about the possibility of surrogate decision-making about participating in research and sharing health information [35]. For example, it is possible that a participant would want their health information to be used for research on a condition that they have or may affect members of their extended family; these types of preferences should be expressed to providers as well as to surrogate decision-makers. Participants who demonstrate altruistic intentions when considering sharing their information should be robustly informed of the implications of doing so. A participant who is altruistic in general or is even altruistic about participating in a research study may not be comfortable or feel the same sense of obligation to allow their information to be used beyond the scope of that particular study. There is a need for future research to better understand the role of mechanisms explored in this study—including the experience of discrimination, trust, and preferences about health information sharing. Qualitative studies may better capture some of the compromises participants anticipate when they think about allowing their health information to be used. This type of data can help us to develop policies and practices that address participants' concerns while also tailoring policies to consider the intersection of lived experiences, concerned and preferences, and technological advancement.

Finally, while research subjects may not necessarily expect compensation for their health information, there may be opportunities to improve the process by which participants are acknowledged or recognized. Unlike the donation of an organ, where the donor is assured that their organ will be given to a recipient in need, or donating money to a charitable cause, where the donor is often recognized by name, participating in research has a less direct and observable effect. A research participant's generosity may never be individually recognized and their contribution may not be observed or even realized for decades, if ever. For example, the electronic health record could show participants the different types of studies their health information has contributed to; it could even illustrate the number of patients who were treated for a particular condition because of a participant's health information. These types of changes would require consideration of confidentiality (i.e., ensuring that the participant's personal information is not shared with those who have benefited and vice versa) as well as robust

informed consent processes to ensure that participants do not feel coerced or compelled to participate, for instance, for desirability.

## Limitations

There are some limitations of this study. First, participants were asked about altruism using a series of items that captured insight into their beliefs about different types of behaviors and attitudes about generosity and selflessness. In contrast, the item measuring health information use asked about an ethical obligation; it is possible that the phrase "ethical obligation" was interpreted as being more invasive or intrusive than one's own decision to behave altruistically. Nevertheless, the fact that perceived ethical obligation was significantly lower across all characteristics suggests that participants resist the notion that permission to share health information is an ethical obligation at all. Second, our classification of the ethical obligation variable resulted in analyses that can tell us about the differences between people who do, and do not, believe there is an ethical obligation to allow health information to be used for research, rather than information about degrees of commitment to the belief that there is an ethical obligation. Third, our study draws exclusively on cross-sectional survey data limiting our ability to make causal inferences or understand the role of participants' motivations and concerns more deeply. Future studies using qualitative and quantitative approaches would provide this critical insight.

The health system depends on subject participation in research in order to advance knowledge and subsequently develop treatments, practices, and improve quality of care. There is a presumption that people are generous and will continue to participate in research. However, with the changing nature of participation facilitated by advances in different types of technologies, the system has reacted to failures in protecting research subjects rather than proactively changing policies and practices to build trust and minimize perceived or experienced undue risks and burdens. Most striking is our finding that in a U.S. nationally representative sample the mean altruism score is higher than the mean score for the ethical obligation to allow health information to be used for research. There is a need to examine the mechanisms through which experiences, perceptions, concerns, and preferences influence the belief that people have an ethical obligation to allow health information to be used for research.

## Supporting information

**S1 Data.**
(DTA)

**S2 Data.**
(XLSX)

**S1 File.**
(PDF)

## Acknowledgments

We thank members of the Lifecycle of Health Data Team for their feedback on the manuscript.

## Author Contributions

**Conceptualization:** Minakshi Raj, Raymond De Vries, Sharon L. R. Kardia, Jodyn E. Platt.

**Formal analysis:** Minakshi Raj.

**Funding acquisition:** Sharon L. R. Kardia, Jodyn E. Platt.

**Investigation:** Sharon L. R. Kardia.

**Methodology:** Minakshi Raj, Paige Nong, Sharon L. R. Kardia, Jodyn E. Platt.

**Project administration:** Paige Nong.

**Resources:** Jodyn E. Platt.

**Supervision:** Raymond De Vries, Sharon L. R. Kardia, Jodyn E. Platt.

**Visualization:** Minakshi Raj.

**Writing – original draft:** Minakshi Raj, Jodyn E. Platt.

**Writing – review & editing:** Minakshi Raj, Raymond De Vries, Paige Nong, Sharon L. R. Kardia, Jodyn E. Platt.

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
