## [Decision Letter · Decision Letter 0]

27 Nov 2020

PONE-D-20-29473

Do people have an ethical obligation to share their health information? Comparing narratives of altruism and health information sharing in a nationally representative sample

PLOS ONE

Dear Dr. Raj,

Thank you for submitting your manuscript to PLOS ONE. After careful consideration, we feel that it has merit but does not fully meet PLOS ONE’s publication criteria as it currently stands. Therefore, we invite you to submit a revised version of the manuscript that addresses the points raised during the review process.

We look forward to receiving your revised manuscript.

Kind regards,

Madhavi Bhargava, MD

Academic Editor

PLOS ONE

Additional Editor Comments:

One reviewer has accepted the manuscript with few reservations about the depth and possible conclusions. You may consider including a section on limitations, implications and recommendations. Another reviewer has suggested minor edits. 

Journal Requirements:

Reviewers' comments:

Reviewer's Responses to Questions

**Comments to the Author**

1. Is the manuscript technically sound, and do the data support the conclusions?

Reviewer #1: Yes

Reviewer #2: Partly

2. Has the statistical analysis been performed appropriately and rigorously? 

Reviewer #1: Yes

Reviewer #2: N/A

3. Have the authors made all data underlying the findings in their manuscript fully available?

Reviewer #1: Yes

Reviewer #2: No

4. Is the manuscript presented in an intelligible fashion and written in standard English?

Reviewer #1: Yes

Reviewer #2: No

5. Review Comments to the Author

Reviewer #1: This is a piece within the complex area of consent and research participation that explores the motivation of research participants and the response to sharing health data for research, even as our health data is less secure than we imagine, and it is not possible to address our common health concerns using data analysis, AI and computation in a world of information networking and meshing.

As there is no qualitative element, it is hard to explore the motivations and concerns in more detail. However this initial study, while bearing limitations in scope and application, can be a springboard for further deeper exploration and research in this space.

Since there were many demographic variables, (possibly mirroring the population under study) it would be hard to arrive at information that can effectively inform policy. The depth of understanding is missing and the cross analysis leaves one in a further quandary, which one hopes will be the basis of a further, and better, study that would better elicit these perceptions. Hybrid studies are on way to improve understanding of the data collected

Reviewer #2: The topic is interesting and the sample size is good.

However, the language needs improvement.

The presentation of tables could be in line with the journal guidelines.

Please go through the attachment for an additional comment.

6. PLOS authors have the option to publish the peer review history of their article (what does this mean?). If published, this will include your full peer review and any attached files.

Reviewer #1: No

Reviewer #2: **Yes: **Dr Animesh Jain

---

## [Author Response · Author response to Decision Letter 0]

14 Dec 2020

Editor comments

1. One reviewer has accepted the manuscript with few reservations about the depth and possible conclusions. You may consider including a section on limitations, implications and recommendations. Another reviewer has suggested minor edits. 

We have extended the Limitations section to include some of the concerns Reviewer 1 brought up, on page 24:

Third, our study draws exclusively on cross-sectional survey data limiting our ability to make causal inferences or understand the role of participants’ motivations and concerns more deeply. Future studies using qualitative and quantitative approaches would provide this critical insight.

We provide several recommendations from pages 22-24; however, on page 23 we have added:

This type of data can help us to develop policies and practices that address participants’ concerns while also tailoring policies to consider the intersection of lived experiences, concerned and preferences, and technological advancement.

Reviewer #1: This is a piece within the complex area of consent and research participation that explores the motivation of research participants and the response to sharing health data for research, even as our health data is less secure than we imagine, and it is not possible to address our common health concerns using data analysis, AI and computation in a world of information networking and meshing.

As there is no qualitative element, it is hard to explore the motivations and concerns in more detail. However this initial study, while bearing limitations in scope and application, can be a springboard for further deeper exploration and research in this space.

Since there were many demographic variables, (possibly mirroring the population under study) it would be hard to arrive at information that can effectively inform policy. The depth of understanding is missing and the cross analysis leaves one in a further quandary, which one hopes will be the basis of a further, and better, study that would better elicit these perceptions. Hybrid studies are on way to improve understanding of the data collected. Thank you for your supportive comments on our manuscript. We agree that there is a need to explore this complex area further using other approaches (e.g., qualitative methods) to capture some of the complexity underlying our quantitative findings. Our findings also guide further analyses focusing on specific variables that may be particularly relevant for policy.

Reviewer #2: The topic is interesting and the sample size is good. However, the language needs improvement. The presentation of tables could be in line with the journal guidelines. Please go through the attachment for an additional comment. Thank you for your suggestions. We have made the following changes: 

- On page 4, deleted “that research”

- On page 7, edited to “looking at”

- We are not sure what aspect of the tables is not aligned with journal guidelines. However, we have included gridlines and ensured that text justification is consistent. Our preference is to have the first column (variable names) right-justified, and subsequent columns (data) left-justified.

Wouldn’t confidentiality be threatened in such a case? Would the participants become more vulnerable? Will this lead them to participate to be viewed as positive to the idea? Your point about people who might participate because they want to be viewed positively is particularly interesting. On page 24, we have added:

These types of changes would require consideration of confidentiality (i.e., ensuring that the participant’s personal information is not shared with those who have benefited and vice versa) as well as robust informed consent processes to ensure that participants do not feel coerced or compelled to participate, for instance, for desirability.

Journal requirements 

We confirm that our file names for the revision meet PLOS ONE’s style requirements.

We have attached the whole survey as Supporting Information.

We have included the data relevant for this analysis and a data dictionary as Supporting Information.

---

## [Editor Report · Decision Letter 1]

16 Dec 2020

Do people have an ethical obligation to share their health information? Comparing narratives of altruism and health information sharing in a nationally representative sample

PONE-D-20-29473R1

Dear Dr. Raj,

We’re pleased to inform you that your manuscript has been judged scientifically suitable for publication and will be formally accepted for publication once it meets all outstanding technical requirements.

Kind regards,

Madhavi Bhargava, MD

Academic Editor

PLOS ONE

---

## [Editor Report · Acceptance letter]

21 Dec 2020

PONE-D-20-29473R1 

Do people have an ethical obligation to share their health information? Comparing narratives of altruism and health information sharing in a nationally representative sample 

Dear Dr. Raj:

I'm pleased to inform you that your manuscript has been deemed suitable for publication in PLOS ONE. Congratulations! Your manuscript is now with our production department. 

Kind regards, 

on behalf of

Dr. Madhavi Bhargava 

Academic Editor

PLOS ONE